# Airway Management during Massive Gastric Regurgitation Using VieScope or Macintosh Laryngoscope—A Randomized, Controlled Simulation Trial

**DOI:** 10.3390/jcm11185363

**Published:** 2022-09-13

**Authors:** Hannes Ecker, Sebastian Stranz, Simone Kolvenbach, Holger Herff, Martin Hellmich, Wolfgang A. Wetsch

**Affiliations:** 1University of Cologne, Faculty of Medicine, and University Hospital Cologne, Department of Anesthesiology and Intensive Care Medicine, Kerpener Str. 62, 50937 Cologne, Germany; 2University of Cologne, Faculty of Medicine, and University Hospital Cologne, Institute of Medical Statistics and Computational Biology, Kerpener Str. 62, 50937 Cologne, Germany

**Keywords:** endotracheal intubation, pulmonary aspiration, regurgitation, laryngoscope, VieScope

## Abstract

In this model of massive gastric aspiration, we compared two different laryngoscopes (VieScope and Macintosh) in a randomized, controlled simulation study. The primary endpoint was time to intubation; the secondary endpoints were intubation success (i.e., tracheal tube position) and amount of pulmonary aspiration. Thirty-four anesthetists performed endotracheal intubation using VieScope and Macintosh laryngoscopy in a randomized order on an airway manikin simulating massive regurgitation of gastric fluid. The primary endpoint “time until intubation” could be achieved significantly faster (mean −12.4 s [95% confidence intervals (CI) −19.7 s; −7.3 s]) with Macintosh compared to VieScope (*p* < 0.001). Concerning “correct tube position”, no statistical difference was found between the devices (*p* = 1.0). The mean time to first ventilation was −11.1 s [95% CI −18.3 s; −5.3 s] when using Macintosh (*p* = 0.001). The mean volume of aspirated gastric fluid was lower in the Macintosh group: −90.0 mL [95% CI −235.0 mL; −27.5 mL] (*p* = 0.011). Data from this simulation study suggest that in a model of massive gastric regurgitation, airway management can be achieved faster and with less gastric aspiration when using a Macintosh laryngoscope compared to a VieScope laryngoscope.

## 1. Introduction

Severe pulmonary aspiration is often caused by the entry of gastric contents into the pulmonary system. Consequences can range from pulmonary injury to asphyxial death by obstruction within minutes. Aspiration is most likely when two conditions occur at the same time. The first condition is insufficiency of the lower esophageal sphincter, which can be caused by a decrease in overall muscular tonus (e.g., all conditions that lead to decreased cerebral perfusion), insufficiency of the lower esophageal sphincter (e.g., gastroesophageal reflux disease, hiatus hernia, etc.) or due to an increased abdominal pressure (e.g., postprandial filled stomach, obesity, pregnancy, ileus disease, gastrointestinal bleeding etc.) that facilitates a retrograde flow of gastric content [1,2,3,4]. The second condition that promotes severe aspiration is the loss of protective reflexes, which can be either pathological (this includes all states of reduced consciousness) or iatrogenic (most often caused by the induction of general anesthesia). The combination of these circumstances is most often found during planned (induction of general anesthesia) or unplanned (emergency) airway management procedures. 

Therefore, it is not surprising that anesthetists and emergency physicians must most regularly deal with patients at risk of severe aspiration during airway management in the induction of general anesthesia or in emergency situations [5]. 

In patients who are at risk, the gold standard for airway management in regurgitating patients is endotracheal intubation by rapid sequence induction (RSI) to minimize the risk of (further) aspiration [5,6]. Endotracheal intubation is usually achieved by direct laryngoscopy (most often by using laryngoscopes with the curved Macintosh blade, which is still considered the gold standard in airway management). In the last decade, indirect technologies (“video laryngoscopes”) that facilitate an indirect view of the glottis have emerged and tend to be used, especially for difficult airway management. However, their role in RSI has not yet superseded the use of direct laryngoscopy. 

Especially in cases of severe regurgitation, securing the airway can become extremely difficult when fluids or particles are blocking the sight and may also obstruct the airway itself. Aspiration is a well-known risk factor for failed airway management, as well as a severe contributor to anesthesia-associated morbidity and mortality [7]. Thus, when introducing new intubation devices, handling and safety must be evaluated before they are introduced into clinical practice.

The new Miller-shaped VieScope (Adroit Surgical LCCC, Oklahoma City, OK, USA) is a direct laryngoscope with light transmission through a closed circular tube “to give the user the best illumination of the target tissue with minimal chance of light obstruction by secretions or blood” [8]. However, this statement has not yet been scientifically proven by published data. Therefore, we compared the novel VieScope with a Macintosh laryngoscope (being standard in clinical care in our hospital) in a model of severe gastric regurgitation in a randomized, controlled simulation trial. We formally hypothesized that the VieScope is comparable to Macintosh laryngoscopy in a patient with massive gastric aspiration.

## 2. Materials and Methods

### 2.1. Ethics Approval and Study Registration

The Ethics Committee of the University of Cologne approved the study on October 26, 2020 (ID 20-1465_1; Head: Prof. Dr. Drzezga).

The study was registered at the German Clinical Trials Register, www.drks.de (Identifier: DRKS00023412), on 22 October 2020.

This manuscript is written in accordance with CONSORT guidelines.

### 2.2. Materials

For this study, we compared performance of a conventional Macintosh laryngoscope blade (which was used as standard device for airway management in the hospital) with the novel VieScope (VieScope “Training Demo”, Adult Size, Adroit Surgical LCCC, Oklahoma City, OK, USA). The latter consists of a transparent circular straight tube (comparable to a Miller laryngoscope blade), which is illuminated by LEDs, and a battery handle. The VieScope is a standalone device, battery powered, and disposable after a single use (Figure 1).

As the scope itself has a straight, Miller-shaped tube blade, it facilitates a direct and straight view of the glottis through a transparent and illuminated plastic tube, but it does not allow direct intubation due to the relatively small tube diameter. Instead, it requires the insertion of a bougie once sight to the vocal cords is achieved. Afterwards, an endotracheal tube can be passed into the trachea over the bougie, which then can be removed. For the bougie, the VOIR Tactical Bougie (Adroit Surgical LCCC, Oklahoma City, OK, USA) was used in the attempts with VieScope, whereas a conventional stylet (Portex Stylet medium, Smiths Medical, Ashford Kent, GB) was offered for use in conventional laryngoscopy with the Macintosh blade at the users’ discretion, but was not actually used by any participant.

To avoid friction in the manikin’s airway when using the equipment, every part was lubricated with technical silicone lubricant before the beginning of each trial run.

### 2.3. Study Design

This study was conducted from 19 March to 12 April 2021, in the facilities of the University Hospital of Cologne, as a randomized, controlled manikin trial. Thirty-five physicians, all physicians working at the Department of Anesthesiology and Intensive Care Medicine at the University Hospital of Cologne, volunteered to participate after giving written and informed consent. All participants had received a standardized training with VieScope and had used the device in an airway manikin (>30 min), but the device had not been implemented in routine clinical practice yet.

The inclusion criteria were individuals working as physician in anesthesia or critical care, and aged between 25 and 65 years. There were no exclusion criteria.

### 2.4. Study Protocol

After informed consent, the following demographic and medical background data of the test participants were recorded in pseudonymized form:Gender;Age;Specialization;Medical experience level (years of professional experience);Approximately how many intubations performed per year.

The participants were asked to perform endotracheal intubation on a certified airway training manikin (AirSim Advance X, TruCorp Ltd., Lurgan, Northern Ireland, UK) placed horizontally on an operating table, using VieScope (Adroit Surgical LCCC, Oklahoma City, OK, USA) and conventional laryngoscopy (Macintosh-Blade Size 3, Heine, Herrsching, Germany) in a randomized order.

To simulate massive regurgitation of gastric content, the esophagus of the training manikin was retrofitted with a stomach tube (Dahlhausen Stomach tube, Size Ch.36, Diameter 12.0 mm; P. J. Dahlhausen & Co., Cologne, Germany) inserted from the gastric side, with its tip ending in the upper esophagus approximately 5 cm below the laryngeal level. This tube was connected to a 1.5 Liter canister (Fresenius Hydrobag, Fresenius, Bad Homburg v. d. H., Germany) filled with a brown, highly opaque fluid (i.e., coffee) simulating gastric content (Figure 2).

Coffee was deliberately chosen because of its opacity, thus not allowing any sight of the larynx through the liquid, which is not given in many commercially available fluids such as theatre blood. The canister was placed on a drip stand at a predefined level (100 cm above the manikin’s head) to allow for high flow rates. Participants were provided with a vacuum suction device (Laerdal LCSU 4 suction pump; Laerdal, Stavanger, Norway) for clearing the airway, if needed. Then, for the beginning of each scenario, the simulated regurgitation was started with a flow of approximately 400 mL/min.

To quantify the amount of pulmonary aspirate, left and right lung of the manikin were replaced by, and separately connected to, suction bags to collect all fluid entering the trachea that was not aspirated with the suction pump by the participants.

Each participant used both devices in a randomized order in a crossover design. The order in which the devices were used was randomized using sealed opaque envelopes. A blocked randomization strategy was generated using an online tool (Sealed Envelope Ltd. 2020: www.sealedenvelope.com/simple-randomiser/v1/lists (accessed on 6 October 2020)).

Time measurements started with the beginning of airway measures (taking up the laryngoscope) and ended with the initial correct ventilation (using a resuscitation bag). The volume of the aspirated fluid was measured by weighing both suction bags with a letter scale (where tare was set to the empty weight of the bags). The following data were recorded in a pseudonymized manner for all simulations:Time until intubation (primary endpoint);Tube position: tracheal vs. esophageal (secondary endpoint);Time until first correct ventilation (secondary endpoint);Handling time/Time until bougie (secondary endpoint);Pulmonary aspirated volume (secondary endpoint).

Each simulation was terminated after successful intubation or after 5 min, at which irreversible hypoxia of a real patient would have to be assumed.

### 2.5. Statistical Analysis

Statistical analysis was performed by the Institute of Medical Statistics and Computational Biology of the University of Cologne. Statistical computations were performed using SPSS Statistics (Version 28; IBM Inc., Armonk, NY, USA).

Sample size calculation with estimated intubation times of 20 ± 5 s with Macintosh vs. 25 s with VieScope (enrolment ratio 1:1; α = 0.05; β = 0.20) revealed that 32 participants were required in order to achieve 80% power. Thus, we decided to include 35 participants to compensate for possible dropouts.

For the comparison of “tube position” by Macintosh and VieScope, Fisher’s exact test was performed; “time to intubation”, “time to ventilation”, “handling-time”, and “pulmonary aspirated volume” were tested for normal distribution using Shapiro–Wilk test. Wilcoxon signed-rank test for related samples was used to test the null hypothesis for data that were not normally distributed. Median of the differences and confidence intervals (CI) were calculated using Hodges–Lehman median difference for related samples. A *p* value of <0.05 was considered significant.

## 3. Results

### 3.1. Demographic and Background Data

Thirty-five participants, all staff anesthesiologists, volunteered to participate in this study. There was one dropout during study enrollment, as one participant had to leave the experiment due to urgent clinical duties. Of those remaining 34, 17 were female and 17 were male (Table 1). The mean age was 34 years (range 27 to 46 years). Of all 34 participants, all n = 34 (100%) had experience with Macintosh, while only n = 10 (29%) had prior experience in the use of the VieScope (as this new device was not routinely used in clinical routine at the time of the study). Further characteristics of the participants’ professional experience are given in Table 1.

In randomized order, each participant performed two endotracheal intubation attempts (one with VieScope and one with Macintosh). Thus, 68 data sets were acquired (Figure 3).

### 3.2. Time to Intubation

For the primary endpoint, the time to endotracheal intubation was 34.8 ± 27.6 s (mean ± SD) for Macintosh vs. 49.5 ± 30.9 s for VieScope.

The Wilcoxon signed rank test for related samples revealed that intubation with Macintosh was significantly faster than with VieScope (*p* < 0.001). The Hodges–Lehman median difference was −12.4 s [95% CI: −19.7 s; −7.3 s].

### 3.3. Tube Position-Endotracheal vs. Esophageal

Regarding tube position, n = 32 (94%) achieved a correct endotracheal position with the Macintosh compared to n = 31 (91%) with the VieScope. Statistical analysis revealed no significant difference (*p* = 1.000).

### 3.4. Time to First Ventilation

The time to first ventilation was 44.9 ± 32.2 s for Macintosh vs. 58.3 ± 32.9 s for VieScope. The Wilcoxon signed rank test showed a significant difference (*p* = 0.001). The Hodges–Lehman median difference was −11.1 s for Macintosh [95% CI: −18.3 s; −5.3 s].

### 3.5. Handling Time/Time until Bougie

The time until bougie placement for VieScope was a mean duration of 31.9 ± 23.8 s compared to time complete endotracheal intubation with the Macintosh of 34.8 ± 27.6 s, which was not different in the Wilcoxon signed rank test (*p* = 0.215). The Hodges–Lehman median difference was 2.7 s [95% CI: −2.0 s; +7.6 s].

### 3.6. Pulmonary Aspirated Volume

The pulmonary aspirated volume of gastric fluid was 149.2 ± 141.2 mL with Macintosh, compared to 257.6 ± 269.1 mL for VieScope, indicating significantly less aspirate when using Macintosh (*p* = 0.011) using the Wilcoxon signed rank test. The Hodges–Lehman median difference was −90.0 mL [95% CI: −235.0 mL; −27.5 mL].

### 3.7. Subgroup Analysis—Dependency of the Study Endpoints on Work Experience

A subgroup analysis on the dependency of the study endpoints on work experience is shown in Table 2. Although the study was not adequately powered to detect differences in these subgroups, the Mann–Whitney U test found significant differences between the devices in “Time to intubation” and “Time to ventilation” in the subgroup of participants with 0–3 years’ work experience, but not in the subgroups with more work experience.

## 4. Discussion

This study compared the use of the novel VieScope with the classic Macintosh laryngoscope for airway management in a simulation of massive gastric regurgitation in a randomized, controlled trial. The primary endpoint “time to intubation” could be achieved significantly faster with Macintosh than with VieScope. Concerning correct tube position, no statistical difference was found between the devices. The time to first ventilation was significantly longer for VieScope compared to Macintosh, although there was no difference in handling time. The volume of aspirated gastric fluid was significantly higher in the VieScope group.

This is the first study to investigate the VieScope in a model of gastric aspiration. So far, there are only three published studies using VieScope, all of which are simulation trials focusing on difficult airway management in adults (cervical inline stabilization, tongue edema) or the pediatric airway [9,10,11]. In comparison, the results of this study considering correct tube position, time to intubation, and time to first ventilation confirm data from one of our previous studies, in which we compared VieScope to Macintosh and video laryngoscopy in normal and difficult airway settings [9]. In contrast, a study by a different group—assessing 42 paramedics in a simulation trial—found a shorter time until intubation for VieScope in comparison to Macintosh [11].

Comparing intubation times between VieScope and conventional Macintosh laryngoscopes is somewhat difficult, as VieScope mandatorily requires the introduction of a bougie and does not allow for direct intubation, whereas the use of bougies is facultative when using classic Macintosh laryngoscopes and—depending on local standards—is not common at all in many hospitals. Since the use of bougies is also uncommon in our hospital, we did not make this mandatory for this study, and in fact no participant used a bougie with the Macintosh laryngoscope. The additional step for VieScope is a relative disadvantage for this device in our setting. To counterbalance this, we created the metric of “handling time” (or “time to bougie”) for VieScope and compared it to time until intubation for Macintosh, to obtain information about the timing of the first successful introduction of an airway device in the trachea. There was no significant difference in this handling time, which was also in agreement with the results of our prior study [10]. However, this result is limited in value by the fact that an airway, especially in an aspirating patient, is first secured when the tube is correctly placed, and the cuff is inflated. Furthermore, special bougies may allow oxygen insufflation and thus apneic oxygenation, but no bougie can facilitate real ventilation, which is always the final goal of airway management.

When analyzing the times depending on the users’ work experience, the subgroup analysis revealed significantly slower times to intubation and ventilation in the group of participants with 0–3 years’ work experience. Although this study was not adequately powered to detect significant differences in these subgroups, it cannot be ruled out that there would be more significant differences with more participants. However, in our results, there appears to be a tendency that the less experienced users achieved intubation and ventilation more slowly when using VieScope, whereas this difference becomes increasingly smaller in the groups with more experienced users. Further studies seem necessary to confirm these findings.

The volume of aspirated gastric fluid in this study was significantly higher in the VieScope group. This finding may correlate with the longer time to intubation and the additional procedural steps of handling the tube over a bougie. Although the clinical consequences of this difference cannot be simulated in this model, it is well-known that the severity of Mendelson’s syndrome correlates with the amount of acid within the aspirated gastric content [12,13].

### Limitations

Simulation trials using manikins do have several restrictions, as they cannot encompass the full physiological situation of real patients and human factors, such as stress and cognitive overload. However, emergency situations, such as an aspirating patient, are hardly plannable and are normally only assessable in retrospective evaluations. Consequently, manikin trials are the obvious substitute for standardized testing of new technologies.

Our study population (staff anesthesiologists) was very experienced in endotracheal intubation, and the Macintosh device was used daily in their clinical routine, whereas the VieScope was a novel device. Thus, our findings may not be transferrable to all kinds of users (i.e., paramedics and physicians with less experience in airway management etc.).

## 5. Conclusions

Data from this simulation study suggest that in a model of massive gastric regurgitation, airway management can be achieved more quickly and with less gastric aspiration when using a Macintosh laryngoscope compared to a VieScope laryngoscope.

## Figures and Tables

**Figure 1 jcm-11-05363-f001:**
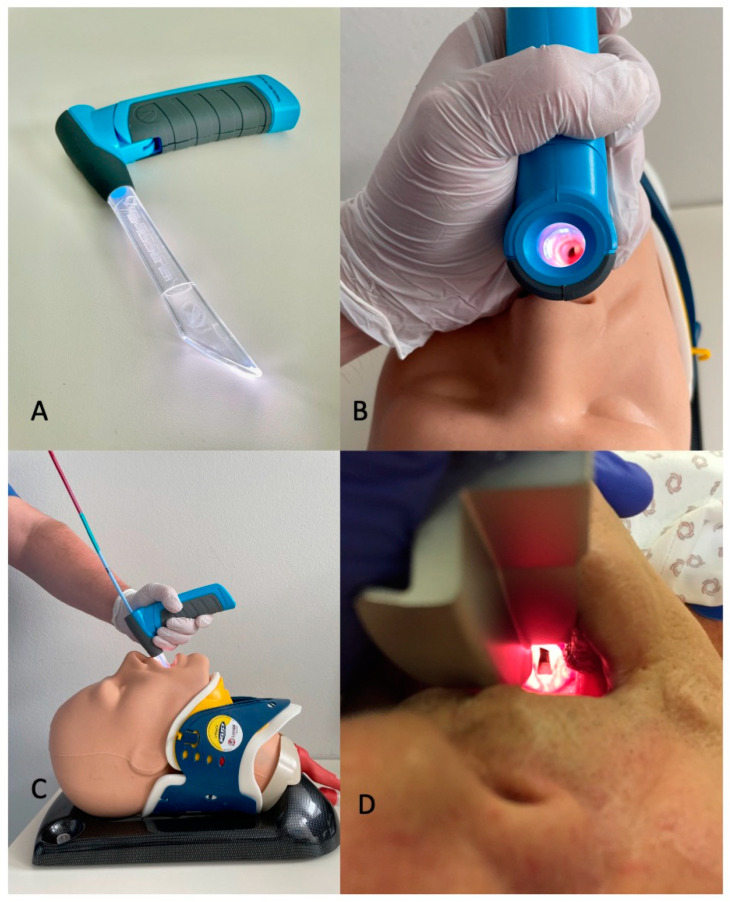
(**A**) VieScope is a novel, Miller-shaped (straight) laryngoscope with circular illumination through an acrylic glass tube. (**B**) It allows visualization of the vocal cords by direct laryngoscopy. (**C**) For introducing an endotracheal tube, the insertion of a bougie is mandatory. After removal of the laryngoscope, the tracheal tube can be passed over the bougie, which remains with its tip in the trachea. (**D**) In contrast, a conventional Macintosh laryngoscope allows direct tracheal intubation after visualization of the vocal cords.

**Figure 2 jcm-11-05363-f002:**
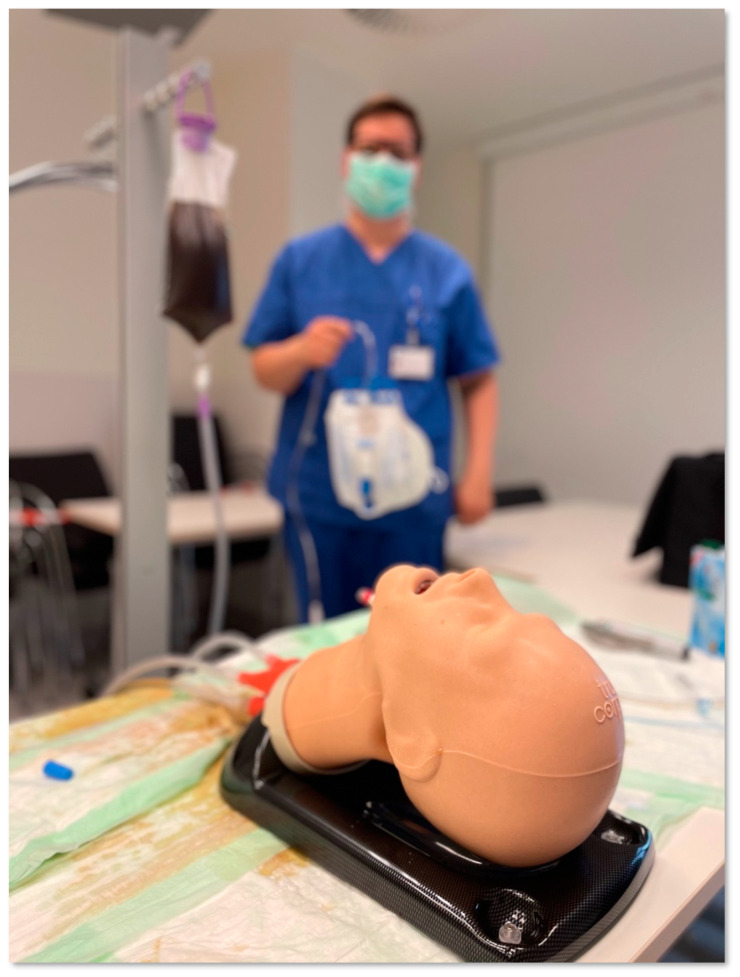
Modified airway manikin. Airway manikin (AirSim Advance X, TruCorp Ltd., Lurgan, Northern Ireland) with a flexible tube ending in the esophagus just below the laryngeal level. This tube is connected to a 1.5 L bag filled with a brown opaque fluid simulating gastric content, which is placed on a drip stand, 100 cm above head level. Both lungs of the manikin were attached to suction bags that could collect regurgitated fluid. (Note: Photograph is taken from a test set-up before actual study conduct.)

**Figure 3 jcm-11-05363-f003:**
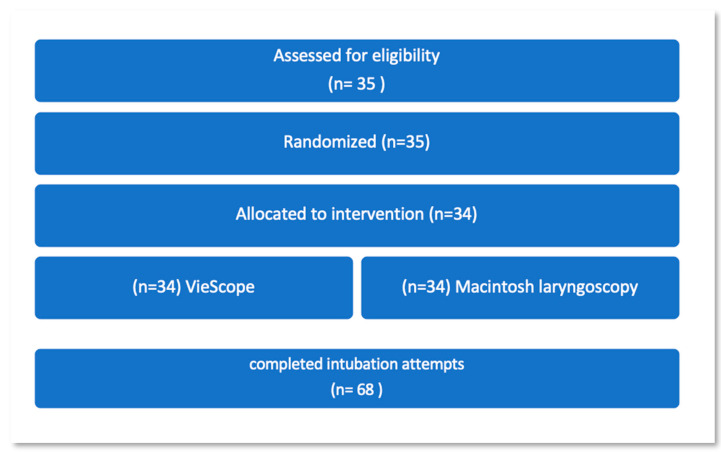
Flow chart. Each participant performed intubation in both settings in randomized controlled order. There was one dropout.

**Table 1 jcm-11-05363-t001:** Professional experience of the participants.

Professional Experience	Experience Level	Number of Participants (n)
work experience	0–3 years	n = 10 (30%)
	3–6 years	n = 9 (26%)
	>6 years	n = 14 (44%)
endotracheal intubations per year	<50 per year	n = 3 (9%)
	50–100 per year	n = 7 (20%)
	100–200 per year	n = 3 (9%)
	>200 per year	n = 21 (62%)

**Table 2 jcm-11-05363-t002:** **Subgroup analysis:** Dependency of the study endpoints on work experience.

Endpoint	Work Experience	VieScope (Median, 25–75%)	Macintosh (Median, 25–75%)	*p*
**Time to bougie ***	0–3 years	33.5 [25.3–51.3]	24.5 [16.5–50.5]	0.364
	3–6 years	24.0 [18.0–43.5]	28.0 [17.5–39.5]	1.000
	>6 years	21.0 [16.0–29.0]	27.0 [20.0–35.0]	0.253
**Time to intubation**	0–3 years	56.0 [40.0–89.5]	24.5 [16.5–50.5]	**0.028**
	3–6 years	40.0 [30.5–80.5]	28.0 [17.5–39.5]	0.122
	>6 years	30.0 [27.0–42.0]	27.0 [20.0–35.0]	0.170
**Time to ventilation**	0–3 years	72.5 [43.5–105.5]	38.0 [22.0–56.0]	**0.032**
	3–6 years	47.0 [36.5–88.5]	37.0 [23.5–46.0]	0.171
	>6 years	42.0 [35.0–53.0]	34.5 [28.8–41.3]	0.111

* Time to bougie compares insertion of bougie in VieScope with intubation in Macintosh; hence, “Time to bougie” and “Time to intubation” are equal in the Macintosh group.

## Data Availability

The original datasets analyzed during the current study are available from the corresponding author on reasonable request.

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
