# Peer review of "Airway Management during Massive Gastric Regurgitation Using VieScope or Macintosh Laryngoscope—A Randomized, Controlled Simulation Trial"

_jcm, 2022, doi:10.3390/jcm11185363_

Round 1

Reviewer 1 Report

This is an interesting mannikin study investigating the use of a novel airway device in the simulated setting of aspiration of gastric contents. The study design and statistical methods are appropriate.

Major comment/s: The paper would benefit by extensive English sub-editing.

Minor comments:

1) Include details of the manufacturer of the Viescope in the introduction.

2) Briefly also mention the nature of the Viescope (potential advantages) and why it was considered worthy of study in this setting (gastric aspiration) in the introduction.

3) In the methods section please bring the description of the materials (Viescope and laryngoscope blade) forward in the description of the methods, instead of leaving it until later in the methods section for the reader to discover what the Viescope is.

4) In the discussion the authors need to justify why the results were not completely predictable given the fact that all the operators were much more experienced using a regular laryngoscope compared to the Viescope and that the Viescope has the additional bougie step - so of course it will take longer and of course there is a higher risk of aspiration.

5) Could the authors also include a statement about the experience the participants had with the Viescope before the study (in regard to comment 4 above)?

6) Could the authors also do a comparison between experienced operators and less-experience operators with regard to the study end-points?

Reviewer 2 Report

SUMMARY

The authors present the results of a randomized controlled simulation trial assessing the efficacy of two different types of laryngoscopes (VieScope and Macintosh laryngoscope) in successful intubation during massive gastric aspiration. Their primary endpoint is time to intubation. Secondary endpoints include intubation success and volume of pulmonary aspiration during intubation. The authors have prepared a well written with clear results presented with appropriate usage of statistical tools.

The manuscript in its current form has a few major issues that the authors need to address.

MAJOR ISSUES

Study design issues:-

            There are 2 major issues with the study design, one of which the authors have addressed in their limitation section.

First, with regards to the limited experience with VieScope of 71% of the participants. Ideally, in such trials for assessing performance, both cohorts must be equal in every other aspect. Since the level of training and experience of one of the device cohorts is so significantly different (71% without experience with VieScope), the two cohorts are not equally leveled for comparison. This difference in level of training is highly likely to bias the performance results. The authors need to clarify if the participants were given any training and/or practice before the trial with VieScope.

Second, both devices have different protocols for intubation. VieScope has an additional step of inserting a bougie before inserting the endotracheal tube. The authors state that they provided the option to use stylet with the Mac laryngoscope to the participants. The authors need to report the participants that utilized the stylet. Both these sub-groups might need to be treated differently. The additional protocol step will further bias the primary endpoint result.      

Motivation for utilizing novel instrument:-

The authors started with the hypothesis that the VieScope’s performance is comparable to Macintosh laryngoscopy in patients with massive gastric aspiration. The authors need to state the motivation for utilization of the new instrument and technique. The authors should discuss the advantages of utilizing the novel instrument.

 Introduction:-

Authors describe the significance and clinical aspects of pulmonary aspiration appropriately. The authors also need to include a summary of current state-of-art techniques and instruments utilized for successful intubation. It can also include upcoming or promising techniques and comment on general trend of this field. Limitations or gaps of these techniques and tools would also help the reader understand the significance and motivation of this study.

Further analysis that might have new results:-

Authors did a great job in collecting their cohort’s experience and practice in Table 1. This is actually an important factor in new technology adoption into practice. I would recommend them to use this data for further analysis. Were there any differences in the performance with VieScope based on clinical experience of participants?

Also, were there difference is performance in participants with and without existing experience with VieScope?

MINOR ISSUES

There were some minor grammatical and spelling issues as listed below.

Line 90: “This tube was which connected to a 1.5 Liter canister…”

Line 146: “...und..”

Line 252: “..VieScope was new a novel device.”

Figure 1: While the figure is self-explanatory, I would encourage the authors to either provide a schematic of the setup or label the components of the setup.

Figure 2: This figure has no particular significance in the study and can be removed. Alternately authors can show both the scopes used for the reader’s benefit. 

Round 2

Reviewer 2 Report

The authors have appropriately addressed all suggestions and comments